# The Effect of Psychological Flexibility on Well-Being during the COVID-19 Pandemic: The Multiple Mediation Role of Mindfulness and Coping

**DOI:** 10.3390/bs14070545

**Published:** 2024-06-28

**Authors:** Thiago Paiva, Ana Nunes da Silva, David Dias Neto, Maria Karekla, Angelos P. Kassianos, Andrew Gloster

**Affiliations:** 1Faculdade de Psicologia, Universidade de Lisboa, 1649-004 Lisbon, Portugal; t.monteiropaiva@gmail.com; 2CICPSI—Centro de Investigação Em Ciência Psicológica, 1649-004 Lisbon, Portugal; 3School of Psychology, ISPA—Instituto Universitário, 1149-041 Lisbon, Portugal; dneto@ispa.pt; 4APPsyCI—Applied Psychology Research Center Capabilities & Inclusion, 1149-041 Lisbon, Portugal; 5Department of Psychology, University of Cyprus, 1678 Nicosia, Cyprus; karekla.maria@ucy.ac.cy; 6Department of Nursing, Cyprus University of Technology, 3036 Limassol, Cyprus; angelos.kassianos@cut.ac.cy; 7Division of Clinical Psychology, Faculty of Behavioral Sciences and Psychology, University of Lucerne, 6002 Lucerne, Switzerland; andrew.gloster@unilu.ch

**Keywords:** psychological flexibility, mindfulness, coping, mental health, well-being, adversity, COVID-19, acceptance and commitment therapy

## Abstract

The COVID-19 pandemic involved a complex set of stressors affecting the health and well-being of the population. The understanding of the psychological processes that influence well-being in response to the pandemic and their interrelation is vital. A promising process in understanding the emotional impacts of these stressors is psychological flexibility. This study investigated the effect of psychological flexibility on well-being, both directly and indirectly, via mindfulness and coping strategies. A total of 334 Portuguese adults participated in this cross-sectional study. Participants were asked to complete an online questionnaire, including measures of psychological flexibility, well-being, mindfulness, and coping strategies. A multiple mediation model studied whether there are conditional indirect effects of mindfulness and coping strategies on well-being. The results showed statistically significant correlations between psychological flexibility, well-being, mindfulness, and coping strategies. Furthermore, we found that mindfulness and coping strategies partly mediate the relationship between psychological flexibility and well-being. Understanding the processes through which psychological flexibility influences well-being in a stressful context is important in comprehending the processes involved in an emotional reaction to a pandemic-like societal event. Mindfulness and coping are shown to be important processes to consider in understanding this phenomenon and designing future responses.

## 1. Introduction

### 1.1. Impacts of COVID-19

The COVID-19 pandemic affected billions of people over a single three-year period. It included stressors inherent to the condition (e.g., death of relatives, being ill), economic impacts, and effects of the lockdown periods [1]. As a global event, it affected entire communities, having a differential impact according to the psychological processes involved. Like other major life events, the COVID-19 pandemic resulted in higher levels of stress, anxiety, and depression [2,3]. The differential impacts may be due to the interaction between personal characteristics and contextual factors [4], and comprehending the psychological processes involved in this interaction helps to further explain these differential impacts. Among these processes, psychological flexibility plays a promising role.

### 1.2. Psychological Flexibility and Well Being

Hayes et al. [5,6] proposed an empirically-based theoretical model under the scope of ACT that operationalised psychological flexibility as the tendency to be aware of internal and external stimuli in the present moment and taking into account the context, responding in ways that accept, persist, change, or facilitate valued goal pursuit. Therefore, it is not considered to be a self-regulatory affective or coping mechanism in itself, but rather a structural mechanism that allows for the selection of the appropriate strategy to be employed according to one’s own set of values. Implied in this selection is a process of adaptation, specifically within adverse situations with unstable and ambiguous stimuli, where the perspective shift is aligned with a mental reconfiguration [7].

In general, holding an attitude of acceptance and openness towards a variety of experiences, as well as willfully choosing to engage with them without avoidance, promotes a rich and meaningful life [5,7]. Inflexibility has been consistently associated with psychopathology, namely depression, stress disorders, anxiety disorders, substance abuse, and eating disorders, among others [8]. Studies have shown that an increased level of psychological flexibility is significantly associated with reduced distress and mental illness and increased well-being [7,8,9,10,11]. Psychological flexibility seemed to have buffered the negative impact of stressors on mental health and well-being [12].

In the context of COVID-19, psychological flexibility seemed to moderate the negative effects of the pandemic on mental health [13]. Additionally, psychological inflexibility mediated the relationship between stress and psychological problems [14]. These findings suggest that, under complex adverse situations, such as those presented by COVID-19 (e.g., quarantine, lack of social contact, and perceived stress), psychological flexibility could help the population to navigate through stressful situations and prevent the emergence of negative psychological outcomes, while promoting mental health and well-being.

### 1.3. Psychological Flexibility and Coping Strategies

A distinct characteristic of psychological flexibility is its potential to be related to how individuals employ different coping strategies in adverse situations. Psychological flexibility itself is not described as a coping mechanism, but rather as a mechanism that contributes to the selection of the type of coping strategy to be recruited [5,6]. Considering this, psychological flexibility is a process that contributes to the type of coping strategies chosen, in which inflexibility would be associated with avoidant coping styles and flexibility with approach coping styles [15,16]. Rueda and Valls [17] have shown through a mediation analysis that psychological inflexibility is linked with maladaptive coping strategies, which, as a consequence, have a negative effect on the symptoms and quality of life of psychiatric patients.

In a study with a sample from the United Kingdom, Dawson and Golijani-Moghaddam [9] found that coping styles partly mediated responses to COVID-19. Avoidant coping strategies were associated positively with anxiety and negatively with well-being, and psychological flexibility was considered a response style that is distinct from the coping strategies themselves [9]. The authors concluded that psychological flexibility acts as a mechanism for predicting how an individual may cope with the pandemic. 

### 1.4. Psychological Flexibility and Mindfulness

Another distinct feature of psychological flexibility is that it aims at promoting contact with the present moment without avoidance. This is achieved via defusion, acceptance, and improving the perspective of the self-as-context, which together form what Chin and Hayes [18] identified as mindfulness processes. Since these processes are embedded in psychological flexibility, it is also of interest to examine whether the effect of psychological flexibility on well-being is attained through the level of an individuals’ mindfulness, which, like coping strategies, will act as a mediator of this relationship. According to Masuda and Tully [11], psychological flexibility and mindfulness are interconnected, but are not redundant and do not substitute or overlap. Wielgus et al. [19] indicated that both psychological flexibility and mindfulness facilitated the achievement of well-being. In contrast, Silberstein et al. [20] asserted that a higher level of both promoted more adaptive dimensions of emotional schemas.

### 1.5. Psychological Flexibility, Well-Being, Mindfulness, and Coping Strategies

While the mechanisms that explain the impact of psychological flexibility on well-being and functioning are not established, mindfulness and coping strategies remain important candidates. Although psychological flexibility relates to the way one copes, it is important to state that it is not what constitutes the coping strategies themselves, but rather how it influences the selection. With mindfulness, the conceptual overlap is more questionable because the concepts come from similar theories. Nevertheless, mindfulness could be seen more as a coping mechanism, while psychological flexibility could be seen as more of a trait that facilitates the use of mindfulness. The COVID-19 pandemic provided an occasion to study these pathways in an emerging and unfolding stressor [21]. Understanding the role of psychological flexibility and the processes by which it operates is vital for developing clinical interventions and public health responses. 

### 1.6. Present Study

Previous studies showed that the variables in our study are interrelated. We propose a multiple mediation model that studies their relationships in an integrated way. By studying psychological flexibility, coping strategies, and mindfulness in a multiple mediation model, we are able to address the issue of how to improve mental health and general well-being through different fronts and perspectives. This is one way to expand and complement the research that has been conducted in this area thus far.

The present study sought to explore the question, “Are coping strategies and mindfulness mechanisms through which psychological flexibility influences well-being?” In answering this question, the study has two goals: first, to conduct a preliminary exploration of the relationship between psychological flexibility, well-being, coping strategies, and mindfulness; and second, to test a double-mediation model regarding this relationship. Two mediators are considered: coping strategies and mindfulness.

## 2. Materials and Methods

### 2.1. Sample

Our sample consists of Portuguese residents from various regions of Portugal, extracted from a larger international sample from the research project COVID_IMPACT, led by the University of Cyprus and the University of Basel [2]. A total of 334 (N = 334) adults participated in the present study; 72.5% were women, and 27.5% were men, aged between 18 and 92 years old, with a mean age of 47.85 (SD = 15.24). In terms of formal education, 43.1% of the participants had a master’s degree, 26.3% had a bachelor’s degree, 13.2% only completed high school, 9% held a PhD, 4.8% attended university without finishing a degree, 0.3% concluded only primary education, and the last 3.3% had another type of educational background. From all the participants, 57.8% worked on a full-time basis, 9.3% worked on a part-time basis, and the others were unemployed (15.3%), retired (15.3%), or on maternity leave (10.8%). A total of 10.8% were studying in institutions of higher education.

### 2.2. Instruments

#### 2.2.1. PsyFlex

In order to measure psychological flexibility, we have used the PsyFlex scale, developed by Gloster et al. [22]. The questionnaire consists of six items that assess psychological flexibility over the last seven days. Each item addresses one of the core skills of the ACT model to promote psychological flexibility and well-being [22]. The questions refer to the experiences of (1) being present, (2) being open for experiences, (3) leaving thoughts be, (4) steady self, (5) awareness of one’s own values, and (6) being engaged, which are then rated on a scale from 5 (“very often”) to 1 (“very rarely”) [22]. A higher score in the total sum represents higher psychological flexibility. The adaptation to Portuguese was performed by Neto et al. [23]. In the present study, PsyFlex had an acceptable internal consistency—α = 0.81.

#### 2.2.2. Mental Health Continuum Short Form (MHC-SF)—Adult Version

Well-being was measured with MHC-SF [24], which assesses three main aspects of well-being: emotional, psychological, and social. This is a multidimensional instrument that produces a total score, as well as separate scores for each of the three aspects of well-being. The present short version includes 14 items. The items are presented on a five-point Likert scale, in which 0 corresponds to “never”, and 5 corresponds to “every day”. The Portuguese version was adapted by Monteiro et al. [25]. The internal consistency of the total scale for our sample is α = 0.90. 

#### 2.2.3. Cognitive and Affective Mindfulness Scale-Revised (CAMS-R) 

The CAMS-R [26] is a 12-item questionnaire that measures four domains of mindfulness: (1) attention, (2) present focus, (3) awareness, and (4) acceptance/non-judgement [26]. These components are consistent with the definitions of mindfulness proposed by Kabat-Zinn [27] and Bishop et al. [28]. The Likert scale in which the items are present varies from 1, “not at all” to 4, “almost always”. A higher score reflects greater mindfulness qualities. We have used the Portuguese version studied by Teixeira et al. [29]. For our sample, Cronbach’s α = 0.85, without item 5. 

#### 2.2.4. Coping Orientation to Problems Experienced Inventory (Brief-COPE)

The Brief COPE [30] has 28 items rated on a 4-point scale, ranging from 1, “I haven’t been doing that at all”, to 4 “I have been doing this a lot”. Eisenberg et al. [31] determined that the various coping strategies reflect mainly two distinct coping styles, i.e., (1) avoidant coping, and (2) approach coping. Avoidant coping comprises self-distraction, self-blaming, substance use, denial, venting, and behavioural disengagement. On the other hand, approach coping comprises acceptance (as in resignation and not actively, as in psychological flexibility), planning, positive reframing, use of emotional support, use of instrumental support, and active coping. Higher scores reflect greater use of avoidant or approach coping styles. We have used the Portuguese version [32], and for our sample, we have obtained a Cronbach’s α = 0.82 for the approach coping items and an α = 0.72 for the avoidance coping items, without items 1 and 19.

### 2.3. Procedure and Data Analysis

After digitally signing an informed consent form, participants were asked to complete a series of online questionnaires regarding the impact of COVID-19 on health [2]. From all the data collected, we have extracted only the variables that are of interest for this study, namely the results of questionnaires that assessed psychological flexibility, well-being, coping strategies, and mindfulness.

In order to statistically analyse our data, we have used IBM SPSS, version 26.0, mainly to generate descriptive statistics, such as means, standard deviations, and frequencies, and to calculate correlations. A Shapiro–Wilk test was conducted in order to assess whether our sample followed a normal distribution. The results showed that our distribution differed significantly from the normal distribution (*p* < 0.05). Hence, non-parametric tests were conducted. 

To test our multiple mediation model, we have used the software PROCESS macro, Framingham, MA, USA [33]. This study followed model 4 for a multiple parallel mediation analysis, in which the effect of our independent variable X (psychological flexibility) on the dependent variable Y (well-being) via the mediators M1, M2, M3 (mindfulness, approach coping, avoidance coping) [33]. Results with a significance level lower or equal to 0.05 are considered statistically significant. 

## 3. Results

### 3.1. Correlation Analysis

Spearman’s correlation was used to analyse the correlation between psychological flexibility, well-being, approach coping, avoidance coping, and mindfulness (see Table 1). The subsequent interpretation of the correlations followed Cohen’s [34] categorisation. In the relationship between psychological flexibility and well-being, we have confirmed our first hypothesis, since a higher level of psychological flexibility is moderately associated with well-being. In terms of the relationship between mindfulness and psychological flexibility, we have found a strong correlation. As for the relationship between mindfulness and well-being, we have found a moderate correlation. 

Regarding the relationship between the level of approach coping and psychological flexibility and well-being, we have found a small correlation between approach coping and psychological flexibility and a moderate correlation between approach coping and well-being. In terms of the relationship between the level of avoidance coping and psychological flexibility and well-being, we have found a small negative correlation between avoidance coping and psychological flexibility (*ρ* = −0.195, *p* < 0.05) and a small negative correlation between avoidance coping and well-being. 

### 3.2. Multiple Parallel Mediation Analysis

Initially, we calculated the direct effect of psychological flexibility on well-being. Our first model, psychological flexibility, emerged as a positive significant predictor of well-being (*β* = 1.92, *SE* = 0.16, *p* < 0.05). Table 2 shows the effects of the mediation model, considering the study’s variables.

The direct effect of psychological flexibility on well-being was positive and statistically significant (*β*1 = 1.02, *SE* = 0.20, *p* < 0.05). The direct effect of mindfulness on well-being was also positive and statistically significant (*β*2 = 0.55, *SE* = 0.14, *p* < 0.05). Regarding the direct effect of approach coping on well-being, we have also found a positive and statistically significant effect (*β*3 = 0.44, *SE* = 0.08, *p* < 0.05). Finally, regarding the effect of avoidance coping on well-being, we have found a negative statistically significant effect (*β*4 = −0.43, *SE* = 0.16, *p* < 0.05). These results confirm our previously mentioned hypothesis.

The indirect effect is statistically significant, leading us to conclude that mindfulness, approach coping, and avoidance coping partially mediate the relationship between psychological flexibility and well-being (Table 3, Figure 1). 

## 4. Discussion

This study examined the effect of psychological flexibility on well-being, considering the mediation role of coping strategies and mindfulness in the context of the COVID-19 pandemic. By investigating the effect of psychological flexibility on well-being through a multiple mediation analysis, we aimed to understand the processes through which psychological flexibility affects well-being. Since the mechanisms that explain the impact of psychological flexibility on well-being and functioning are not established, mindfulness and coping strategies seemed to be important candidates.

Our results showed that psychological flexibility has both a direct effect on the levels of well-being and an indirect effect through mindfulness and coping strategies. The direct effects are according to the existing literature [7,8]. The confirmation in the context of the emotional response to COVID-19-related stressors is an important extension of this finding. By confirming our hypothesis that a higher level of psychological flexibility is correlated to higher levels of well-being, our study is compatible with earlier theoretical assumptions and empirical evidence [5,7,8], allowing us to open new research pathways. 

The two considered mediators complement and further explain this relationship. Mindfulness correlated positively, with a strong association with psychological flexibility and with a moderate association with well-being. The link between mindfulness and psychological flexibility shows us that mindfulness is indeed embedded in psychological flexibility, as suggested by Chin and Hayes [18]. As for the associations with coping strategies, we found small to moderate correlations with approach coping (positive association) and avoidance coping (negative association) regarding both psychological flexibility and well-being. These results support the evidence found in previous studies, specifically in the context of COVID-19, such as the study by McFadden et al. [35], which stated that certain coping strategies found within approach coping (such as active coping and help-seeking) were associated with higher levels of well-being, and some avoidance coping strategies (such as avoidance) were associated with lower levels of well-being.

The consideration of these associations and partial mediations also deepens our understanding of what psychological flexibility is. As previously mentioned, psychological flexibility is considered a structural mechanism that allows for the selection of the appropriate strategy to be employed according to one’s own set of values [9]. This means that a higher level of psychological flexibility indeed affects well-being via a higher level of approach coping and a lower level of avoidance coping. However, the interplay between psychological flexibility and well-being via these two coping styles showed that effective action might be better achieved through an extensive repertoire of coping strategies that include both approach and avoidance behaviours, rather than through solely choosing one style. In fact, our results support the theory proposed by Zimmer-Gembeck et al. [36] that highlights the concept of coping flexibility, in which, depending on the context and the situation, individuals should have the capacity to select from a vast range of coping strategies, choosing the ones that are most suitable for the situation. Psychological flexibility is a concept that encompasses this idea.

Our study contributed to generating more evidence, pointing to the strong relationship between PF and well-being. It also used a new measure of psychological flexibility—Psyflex—operationalised according to the ACT model.

There are also some important limitations in this study. Due to its cross-sectional correlational design, our ability to conclude causality and directionality is limited, and should be cautiously considered. One way to tackle this limitation in further studies would be to use longitudinal designs that test the relationships between our variables over time, further supporting a causal relationship between these factors. The construction of our mediational model was hypothetical and based on the information in the literature. Still, other models could be designed including our, or other, variables (changing mediators, adding moderators, checking for bi-directionality, etc.). To better understand which model better explains the relationship(s) between psychological flexibility and well-being, researchers should statistically compare and test several models in future studies. Some of the components of psychological flexibility are organized into mindfulness processes, which we analyzed in our study by placing mindfulness as a mediator. However, further studies should focus on the other components of psychological flexibility, specifically those organized in valued-based action processes, i.e., identifying core values and promoting effective action to achieve the identified values. In order to compare our results with those from a clinical sample, researchers could examine variables regarding mental illness, such as measures of anxiety and depression, to broaden the application of interventions that target psychological flexibility. Also, the sample size did not allow us to thoughrouly assess the validity of the concepts being applied to the mediation model. The scales used were only validated using simpler methods, such as Chronbach’s alpha, and not by confirmatory factor analysis.

Some clinical implications can be drawn from our study that may help health professionals design interventions that are sensitive to a globally adverse context. On a preventive level, there have been several programs designed to promote psychological flexibility in specific contexts, such as for burnout prevention and the promotion of psychological flexibility in the workplace [37,38], with academic graduate students [39], and for patients with mental illness [40], among others. Nonetheless, our study addressed the question of psychological flexibility and well-being in an adverse global context. For that reason, programs to promote psychological flexibility could be re-designed according to a community-wide perspective. 

Our study showed that mindfulness and coping strategies are important factors in the relationship between psychological flexibility and well-being. Hence, the integration of these components in such programs would be advisable. Extensive empirical evidence for the efficacy of mindfulness programs has been thoroughly presented over the years [41]. This could serve as a starting point to integrate mindfulness, with better use of coping strategies in order to enhance psychological flexibility and achieve well-being in a community-wide intervention. The interplay between psychological flexibility and well-being via coping styles showed that effective action might be better achieved through an extensive repertoire of coping strategies that include both approach and avoidance behaviours, rather than by solely choosing one style. Also, the use of the internet and mass media, as well as digital applications, could be an effective way to reach a broader population. All in all, these recommendations and suggestions aimed at promoting an ongoing insightful analysis of the context of the complexity in which psychological flexibility might play a key role by helping researchers, health professionals, policymakers, and the general population to improve mental health and well-being.

## Figures and Tables

**Figure 1 behavsci-14-00545-f001:**
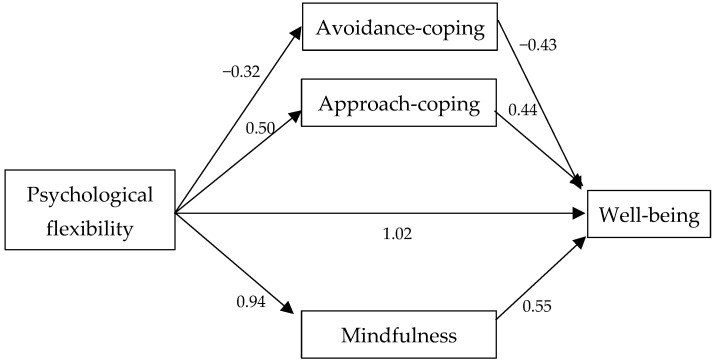
Multiple parallel mediation model.

**Table 1 behavsci-14-00545-t001:** Spearman Correlation between the psychological flexibility, well-being, avoidance coping, approach coping, and mindfulness variables.

	Psychological Flexibility	Well-Being	Avoidance Coping	Approach Coping	Mindfulness
Psychological flexibility	-	0.487 **	−0.195 **	0.233 **	0.609 **
Well-being	0.487 **	-	−0.170 **	0.347 **	0.495 **
Avoidance coping	−0.195 **	−0.170 **	-	0.235 **	−0.256 **
Approach coping	0.233 **	0.347 **	0.235 **	-	0.250 **
Mindfulness	0.609 **	0.495 **	−0.256 **	0.250 **	-

** Correlation is significant at the 0.01 level (two-tailed).

**Table 2 behavsci-14-00545-t002:** Analysis of the effect of the mediation model (model 2).

			Well-Being (R^2^= 0.39; *p* = 0.00)			
	Point Estimates	Confidence Products			95% CI	
	*β*	*SE*	*T*	*p*	Lower Limit	Upper Limit
Psychological flexibility	1.02	0.20	4.98	0.00	0.62	1.43
Mindfulness	0.55	0.14	3.97	0.00	0.28	0.83
Approach coping	0.44	0.08	4.98	0.00	0.27	0.62
Avoidance coping	−0.43	0.16	−2.59	0.00	−0.76	−0.10

**Table 3 behavsci-14-00545-t003:** Bootstrap mediation model: indirect effect.

				Bootstrap Times	5000 95% CI
		*β*	*SE*	Lower Limit	Upper Limit
Mindfulness		0.52	0.13	0.27	0.79
Approach coping	Indirect effect	0.22	0.06	0.10	0.37
Avoidance coping		0.14	0.06	0.03	0.27

*N* = 334; *β* = non-standardized regression coefficient; *SE* = standard-error; CI = confidence interval.

## Data Availability

Data are available upon request.

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
