# Peer review of "The Effect of Psychological Flexibility on Well-Being during the COVID-19 Pandemic: The Multiple Mediation Role of Mindfulness and Coping"

_behavsci, 2024, doi:10.3390/bs14070545_

Round 1

Reviewer 1 Report

Comments and Suggestions for Authors

Thanks for inviting me to review this paper. The paper aims to explore the relationship between psychological flexibility and wellbeing with avoidance coping, approach coping and mindfulness as mediators.

Overall, the paper is well written. My main concern about this paper is related to the theoretical framework of the mediation analyses. The authors have explored PF as a potential predictor of well-being with mindfulness and avoidance as mediators. Psychological flexibility, mindfulness, and avoidance measures potentially include overlapping concepts. Can the authors provide a theoretical rationale and statistical/methodological justification for conducting such mediation analyses (and statistical justification)? Why have they investigated mindfulness and avoidance as potential mediators when these are components of psychological flexibility? 

Can the authors better specify the research questions? They only mention goals and it is not clear what they are interested in exploring 

What is the mean age/age range of the sample size? 

Were participants mainly students?

Comments on the Quality of English Language

Minor edits required but overall is good 

Reviewer 2 Report

Comments and Suggestions for Authors

Dear authors, it was a pleasure reading your paper. I was impressed by the thoroughness of your research and the clarity of your findings. Your study sheds valuable light on the relationship between mindfulness, coping strategies, and well-being among Portuguese adults, providing insights for both researchers and practitioners in the field.

I would like to recommend a slight extension of the conclusion section and to provide a more comprehensive summary of the implications of your findings and potential avenues for future research."For exemple I would reccomand to extend this paragraph Ït could serve as a starting point to integrate mindfulness with better use of coping strategies in order to enhance psychological flexibility and achieve well-being in a community-wide intervention". Additionally, harmonizing the references will enhance the overall coherence of the manuscript. Once again, congratulations for your needed work .

Author Response

Dear Reviewer thank you so much for your review and feedback on this paper.

We tried to address all the issues, mentioned. We tried to extend the conclusion section and provide a more comprehensive summary of the implications of our findings and potential future research. This can be seen highlighted at the discussion section at pages 7 and 8.

We also reviewed the references. Specifically, the self-citations. We excluded previously numbered citation 12

Sebastião, R.; Neto, D.D. Stress and mental health: The role of emotional schemas and psychological flexibility in the context of COVID-19. Journal of Contextual Behavioral Science, 2024, 100736. https://doi.org/10.1016/j.jcbs.2024.100736

since it was not essential to the core comprehension of this research paper.

We maintained the other references considering they were part of previous research that creates a coherent narrative for this paper.

Thank you again for your inputs.

Best regards, 

Reviewer 3 Report

Comments and Suggestions for Authors

well done!

Author Response

Thanks for your review and feedback on this paper.

Best regards, Ana NS